# Microtubule end conversion mediated by motors and diffusing proteins with no intrinsic microtubule end-binding activity

Manas Chakraborty[1,7], Ekaterina V. Tarasovetc[1], Anatoly V. Zaytsev[1], Maxim Godzi[1,2], Ana C. Figueiredo[3,4], Fazly I. Ataullakhanov[2,5,6] & Ekaterina L. Grishchuk [1,5]

Accurate chromosome segregation relies on microtubule end conversion, the ill-understood ability of kinetochores to transit from lateral microtubule attachment to durable association with dynamic microtubule plus-ends. The molecular requirements for this conversion and the underlying biophysical mechanisms are elusive. We reconstituted end conversion in vitro using two kinetochore components: the plus end–directed kinesin CENP-E and microtubule-binding Ndc80 complex, combined on the surface of a microbead. The primary role of CENP-E is to ensure close proximity between Ndc80 complexes and the microtubule plus-end, whereas Ndc80 complexes provide lasting microtubule association by diffusing on the microtubule wall near its tip. Together, these proteins mediate robust plus-end coupling during several rounds of microtubule dynamics, in the absence of any specialized tip-binding or regulatory proteins. Using a Brownian dynamics model, we show that end conversion is an emergent property of multimolecular ensembles of microtubule wall-binding proteins with finely tuned force-dependent motility characteristics.

[1] Department of Physiology, Perelman School of Medicine, University of Pennsylvania, Philadelphia, PA 19104, USA. [2] Center for Theoretical Problems of Physicochemical Pharmacology, Russian Academy of Sciences, 119991 Moscow, Russia. [3] Chromosome Instability & Dynamics Laboratory, Instituto de Biologia Molecular e Celular, Universidade do Porto, Rua Alfredo Allen 208, 4200–135 Porto, Portugal. [4] Instituto de Investigação e Inovação em Saúde – i3S, Universidade do Porto, Rua Alfredo Allen 208, 4200–135 Porto, Portugal. [5] Dmitry Rogachev National Research Center of Pediatric Hematology, Oncology and Immunology, Moscow 117997, Russia. [6] Moscow Institute of Physics and Technology, Dolgoprudny, Moscow Region 141701, Russia. [7] Present address: Centre for Mechanochemical Cell Biology, Warwick Medical School, Coventry CV4 7AL, UK. Correspondence and requests for materials should be addressed to E.L.G. (email: gekate@pennmedicine.upenn.edu)

Accurate chromosome segregation involves kinetochore attachment to dynamic microtubule (MT) plus-ends, which drive chromosome oscillations in metaphase and pull sister chromatids apart in anaphase. Although some kinetochores acquire such end-on attachments via a direct capture of the growing MT tips, most initially interact with the MT walls[1]. The resultant lateral configuration is subsequently converted to MT end-binding either by depolymerization of a distal MT segment or by the transport activity of centromere-associated protein E (CENP-E), a kinetochore-associated plus-end-directed kinesin[2–6]. After the MT plus-end comes in contact with the kinetochore, the chromosome motions become coupled to MT dynamics, as tubulins are added or removed from the kinetochore-embedded MT ends[7–9]. The biophysical mechanisms underlying conversion of lateral attachment into dynamic MT end-coupling are not well understood[10]. For example, it is not known whether such conversion requires proteins with distinct MT-binding activities, i.e., those that interact with the MT wall during lateral CENP-E-dependent transport and those that subsequently bind to the MT tip. The key motility characteristics of these molecular components have not been determined, and it previously remained unclear whether their interactions with the MT must be regulated in order to enable their distinct interactions with MT walls vs. ends.

To investigate these outstanding questions, we used reductionist approaches with stabilized and dynamic MTs in vitro. Specifically, we sought to determine whether the MT wall-to-end transition via the CENP-E-dependent pathway could be recapitulated by combining CENP-E with various kinetochore MT-associated proteins (MAPs). Previous work in cells identified the Ndc80 complex as the major kinetochore MAP responsible for end-on MT coupling[11]. However, past in vitro reconstitutions provided little insight into why this protein plays such a central role given that it has no intrinsic MT end-binding activity. Indeed, single Ndc80 molecules bind to MTs in vitro and undergo transient diffusion along polymerized tubulins in the straight MT wall, but Ndc80 exhibits no strong preference for polymerizing or depolymerizing MT tips[12–15]. Consistent with this, Ndc80 has significantly lower affinity for curved tubulin protofilaments than intact MTs[12,16]. Although antibody-induced Ndc80 clusters[13] and Ndc80-coated microbeads[17] are capable of tracking the dynamic MT ends, this behavior is not unique among MT-binding proteins, including those that have no role in kinetochore-MT coupling[18]. Thus, it remained unclear whether Ndc80 in combination with kinesin CENP-E is capable of supporting MT end-conversion. Here, we show that CENP-E and Ndc80 have finely tuned molecular characteristics enabling them to robustly convert lateral MT attachment into end-coupling in the absence of other kinetochore proteins and regulatory events.

## Results

**Ndc80 exerts molecular friction to CENP-E-driven motility.** Full-length CENP-E can walk to MT plus-ends and briefly (<20 s) maintain MT end-association thanks to a MT-binding domain in its tail region[19]. To determine how Ndc80 effects CENP-E motor motility without the interference of this domain, we used a truncated version of CENP-E that falls off MT tips, but on MT walls it walks and responds to force similarly to the full-length protein[20]. Glass microbeads coated with these motor domains (hereafter referred to as "CENP-E motor") were also coated with purified Ndc80 "Broccoli" complexes containing the wild-type MT-binding domains (Supplementary Fig. 1a). Using a laser trap, we promoted binding of such a bead to the wall of a taxol-stabilized MT lying on a coverslip (Fig. 1a). Subsequent bead's motility depended strongly on the ratio of Ndc80/CENP-E

coating (Fig. 1b). With more Ndc80 present, beads tended to pause at the MT tips, detaching less frequently. However, the beads walked more slowly, and many could not reach the MT plus-ends.

To investigate the origin of the observed decrease in velocity, we turned to the established "gliding assay", in which lateral MT transport is driven by a CENP-E motor sparsely attached to the surface of a coverslip (Fig. 1c). MTs glided at ~20 μm min$^{-1}$ on coverslips coated with CENP-E motor in the Ndc80 absence[21], but velocity decreased when Ndc80 was conjugated to the same coverslips. Importantly, the Ndc80 and CENP-E motor domains have not been reported to physically interact[22]. Moreover, these proteins were immobilized using different antibodies, and soluble proteins were washed away. Therefore, slow velocity was caused via their mechanical coupling to MTs, rather than direct Ndc80-CENP-E binding. The molecular friction generated by binding of Ndc80 to the MT as it glides under the power strokes of CENP-E slows down the motor. This force-dependent effect is specific to Ndc80, as the MT-binding CENP-E tail domain caused only a minor hindrance (CENP-E Tail, Fig. 1c), consistent with the tail's inability to slow down motor domains within the full-length molecule[19].

Importantly, within the optimal range of Ndc80 and CENP-E coatings on the microbeads, the motor was able to overcome this friction and deliver beads to MT plus-ends. On dynamic MTs growing from coverslip-immobilized MT seeds, upon arrival at the tips the beads slowed down even further and continued to move at the normal rate of MT elongation (Fig. 1d, Supplementary Fig. 2). When MTs disassembled, the beads moved backward; these beads detached more frequently than during polymerization, but overall their MT end-coupling was maintained for several minutes.

**CENP-E and Ndc80 enable MT wall-to-end transition.** The results obtained using laser-handled microbeads suggested that a combination of CENP-E motors and Ndc80 molecules can support MT end-conversion. However, this conclusion was tempered by the observation that MAP-coated beads could roll on the MT surface, a highly non-physiological behavior that would disrupt normal MT end-coupling[18]. Furthermore, the thermal motions of the beads at the ends of MT extensions in this assay complicate thorough analysis and visualization of MT-bead coupling. To overcome these limitations, we modified the assay's geometry to use coverslip-immobilized microbeads and freely floating fluorescent MTs (Fig. 2a). As expected, beads coated with Ndc80 bound to the walls, rather than the ends, of GMPCPP (Guanosine-5'-[(α,β)-methyleno]triphosphate)-stabilized MTs. MT walls also bound to CENP-E–coated beads in the presence of AMP-PNP (Adenosine 5'-(β,γ-imido)triphosphate tetralithium salt hydrate), a non-hydrolysable adenosine triphosphate (ATP) analog (Fig. 2b). After ATP was added, the attached MTs glided on the beads (Supplementary Movie 1; Supplementary Figs. 1c and 3). Because CENP-E is a plus-end-directed motor, a MT glides with its minus-end forward, and the MT plus-end is the last contact point between the MT and the bead. Most MTs detached immediately after their plus-ends arrived at the CENP-E-coated bead, while 24% of ends were transiently retained (1.1 ± 0.3 min; Fig. 2c), confirming the inability of CENP-E motor domains alone to form lasting attachments to MT tips[19].

We then supplemented these surfaces with Ndc80 coating that reduced CENP-E gliding velocity fourfold while allowing the majority of MT plus-ends to arrive at the bead. To estimate the number of Ndc80 molecules that interacted with the MT wall under these conditions, we took advantage of the ability of bead-conjugated Ndc80 to support diffusion of the laterally bound MT (in the absence of CENP-E). We then compared this experimental

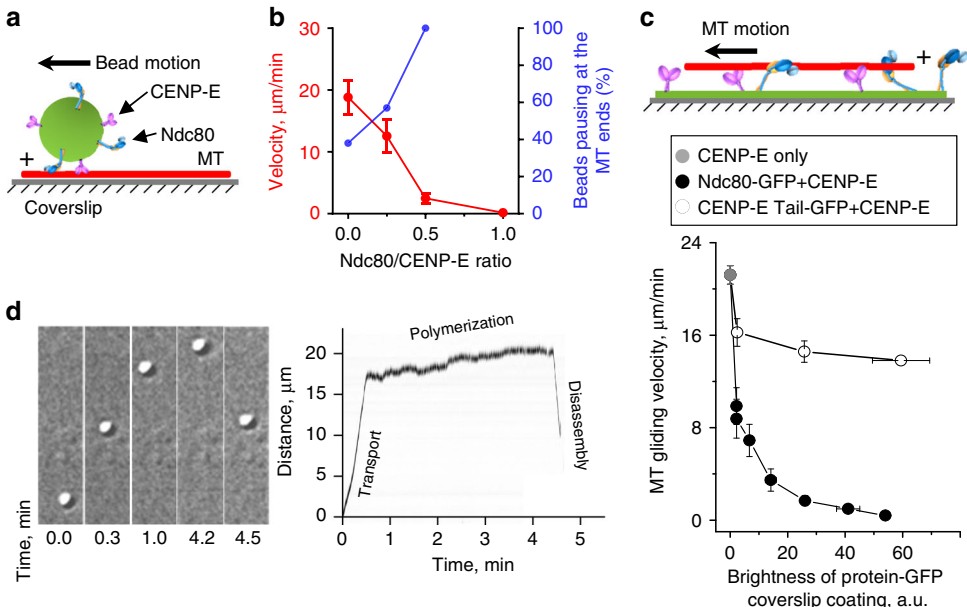

**Fig. 1** Microtubule (MT) wall-and-tip interactions with a molecular lawn of CENP-E motors and Ndc80 complexes. **a** Experiment with taxol-stabilized MT immobilized on a coverslip and a bead with randomly conjugated Ndc80 and CENP-E molecules. **b** Velocity of bead transport along the MT walls (means ± SEM; red curve plotted using left axis) and attachment at the MT end (blue, right axis) versus the ratio of Ndc80 and CENP-E concentrations used for bead coating. Beads were scored as pausing at the MT tip if they remained attached for at least 2 s. Source data are provided as a Source Data file. **c** MT gliding on a coverslip coated with protein mixtures. Top: schematic of coverslip conjugation using tag-binding antibodies, ensuring that MT-binding domains are not sterically inhibited. Bottom: gliding velocity on coverslips with CENP-E motor and either Ndc80-GFP or CENP-E Tail-GFP versus the protein coating density, as determined by green fluorescent protein (GFP) fluorescence. Points are means ± SEM for average velocities from $N = 3$ independent trials, examining total $n$ MTs: for CENP-E $n = 84$; for Ndc80+CENP-E $n = 522$, for Ndc80+CENP-E Tail $n = 305$. Source data are provided as a Source Data file. **d** Example of a bead walking to the end of the dynamic MT at the velocity of CENP-E motor, continuing in the same direction at the velocity of MT polymerization, and then moving backward when the MT disassembles (MT tip tracking). Left: time-lapse images acquired with differential interference contrast. Right: kymograph of the entire trajectory

diffusion rate with the theoretically predicted rate for different numbers of Ndc80 molecules, based on the diffusion rate measured for a single Ndc80 molecule (Supplementary Note 1, Supplementary Fig. 4). We estimate that 11–13 bead-bound Ndc80 molecules engaged in MT binding in our end-conversion assay, similar to the number of Ndc80 molecules interacting with one kinetochore MT[23,24].

Under these conditions, almost 80% of MT plus-ends that arrived at beads containing both proteins remained attached for 18.1 ± 1.2 min (Supplementary Movie 2). The actual end-retention time was even longer, as many end-attachments outlasted a typical experiment. Kaplan–Meier survival analysis, which takes into account the MTs that were still bead-bound at the end of observation, indicated that 80% of MT end-attachments survived for >28 min. On the timescale of mitosis, this constitutes a durable attachment that greatly exceeds results obtained for CENP-E motor alone, with which >80% of MTs detached in <2.5 min (Fig. 2d). Because these beads are immobilized, rolling could be ruled out, and all molecular motions along the MTs must have physiological geometry. These end-attachments clearly relied on Ndc80–MT binding moieties because two Ndc80 mutants that perturb MT interactions (K166D and Δ1–80)[25] failed to maintain end-attachment, whereas the Bonsai mutant, which has a shorter stalk but wild-type MT-binding domains[26], worked well (Fig. 2c). Thus, a combination of only two proteins, CENP-E kinesin and Ndc80, can provide both efficient lateral transport and wall-to-end transition at stabilized MTs.

**Molecular-mechanical model of MT wall-to-end transition**. To increase our confidence in the above conclusions, we used

quantitative methods to recapitulate end-conversion in silico. We employed equations of Brownian dynamics to describe interactions between the MT wall and multiple MT-binding molecules with and without motor activity (see Supplementary Note 1). In the model, these molecules are immobilized on a surface, representing a patch on the microsphere, as in our end-conversion assay (Fig. 3a, Supplementary Fig. 5). They bind and detach from the MT protofilament stochastically and move randomly (Ndc80) or unidirectionally (CENP-E). Because both the walking motors and thermal fluctuations generate force acting on the MT and this force is transmitted to all MT-bound molecules, the velocity of these molecular motions and the unbinding times were modeled as force dependent (Fig. 3b, c; Supplementary Table 1). In this model, as in our experiments, CENP-E brings the MT plus-end to the motor-attachment site. After the last motor walks off the end, the MT detaches from the molecular patch (Supplementary Movie 3). Multiple surface-immobilized molecules of Ndc80, on the other hand, drive lateral MT diffusion, with a rate that decreases as more molecules are modeled (Supplementary Fig. 4). These diffusive MT motions often (for a few Ndc80 molecules), or occasionally (with more molecules), bring one of the MT ends to the molecular patch (Supplementary Movie 4). Because the patch hosts multiple Ndc80 molecules, the probability of complete MT end detachment is very low, even though each Ndc80 molecule binds for only <0.5 s[15,27]. The MT end inevitably diffuses away from the edge, centering on average at the Ndc80 patch, as expected from random diffusion of the wall-binding proteins.

Combining the CENP-E motors and Ndc80 molecules in these simulations revealed highly dynamic and complex ensemble behavior (Supplementary Movie 5). Importantly, the MT only

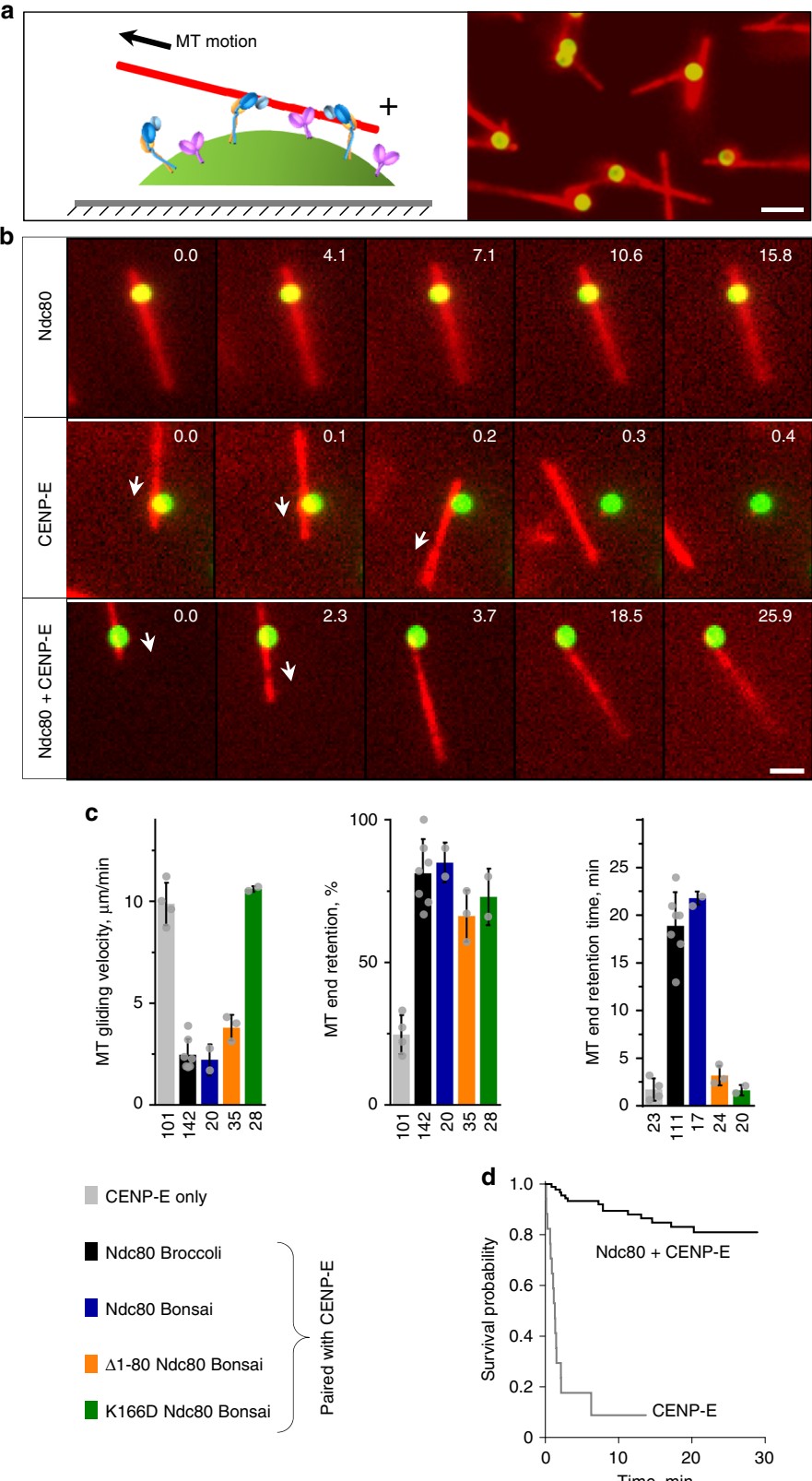

rarely slides all the way to the last motor at the edge of the patch, because as the number of MT-bound motors decreases, they begin to struggle with the MT-bound Ndc80 molecules and frequently dissociate. The MT end, however, does not detach and is even slightly pulled away from the edge by the diffusing Ndc80 molecules. If, however, some or all Ndc80s unbind, the motors resume their persistent transport, trying to decrease the overlap between the MT and the patch. This, in turn, decreases the number of bound motors, the Ndc80/motor ratio increases, the MT slows down, and the cycle begins again. Because multiple molecules are involved, and their stepping and thermal MT fluctuations are stochastic, these phases of the tip motions are

**Fig. 2** Microtubule (MT) wall-to-end transition by CENP-E paired with Ndc80 complex. **a** Schematics of the MT wall-to-end transition assay and a representative imaging field with GMPCPP-stabilized MTs (red) and coverslip-immobilized beads coated with green fluorescent protein (GFP)-labeled Ndc80 protein (green). Bar, 3 μm. **b** Selected images showing motions of MTs on immobilized beads coated with the indicated proteins. Numbers are time (min) from the start of observation. Arrows show direction of MT gliding. Bar, 3 μm. **c** Quantifications for the wall-to-end transition assay using a mixture of CENP-E motors and different Ndc80 proteins. Columns are means ± SD for results from $N$ independent trials, which are shown with gray dots. For CENP-E only, CENP-E paired with either Ndc80 Broccoli, Ndc80 Bonsai, Δ1–80 Ndc80 Bonsai, or K166D Ndc80 Bonsai, $N$ = 4, 7, 2, 3, and 2, respectively. Total number of examined MTs in all trials is indicated below each column. Source data are provided as a Source Data file. A successful MT end-retention event was counted if the trailing MT end was coupled to the bead for longer than 4 s. All Ndc80 proteins showed high percent of end-retention events but their durations were dramatically different. **d** Kaplan–Meier survival plot for MT end-retention time based on $N$ independent trials examining end-attachment for $n$ MTs: CENP-E motor: $N$ = 4, $n$ = 23; Ndc80+CENP-E, $N$ = 7, $n$ = 111

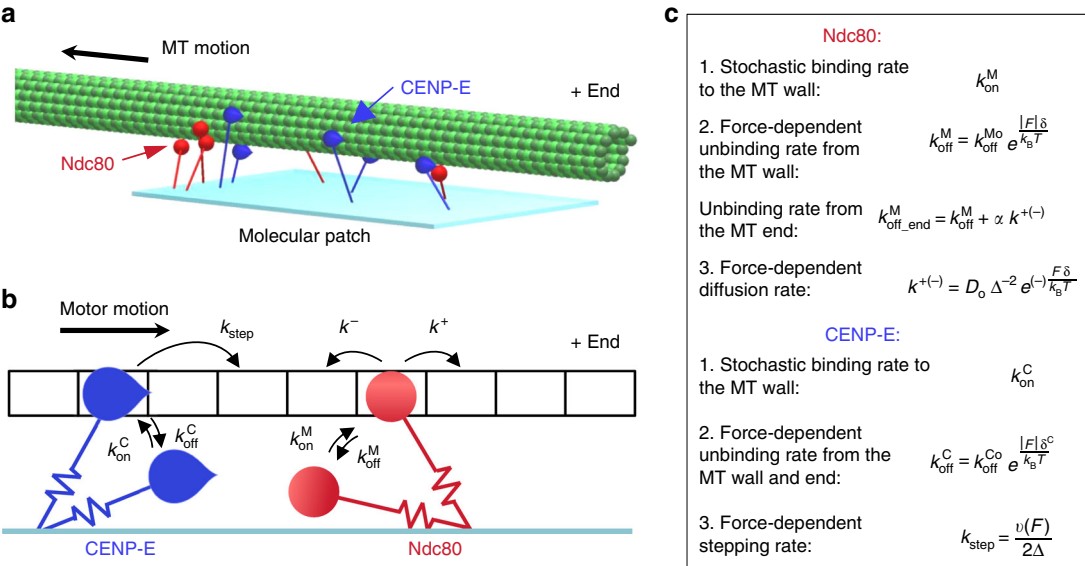

**Fig. 3** Mathematical model of the molecular ensemble of motors and diffusing microtubule-associated proteins (MAPs). **a** Multiple MAPs (red) and motors (blue) are randomly distributed on the surface, forming a molecular lawn. Stabilized microtubule (MT) moves under force from kinesin motors in the presence of thermal noise. **b, c** Summary of kinetic transitions. Molecules bind stochastically to the 4 nm sites on the MT wall, and their unbinding is increased by force. Stepping of the motors and diffusional steps of the MAPs are also force dependent. The motor dissociates from the MT end and the MT wall at the same rate. The MAP molecule can dissociate from the MT end fully or continue to diffuse on the MT wall

highly irregular, and different numbers of molecules are involved at any given time. Importantly, this complex molecular ensemble can maintain small MT overlap (20–40 nm) for a significant time (70% of patch–MT attachments lasted >30 min), recapitulating our findings in vitro. Thus, modeling supports the idea that MT end-conversion can be achieved by proteins with no intrinsic MT end-binding activity.

**The reconstituted end-attachment supports cycles of MT dynamics.** Next, we investigated whether such MT end-attachments could be transformed into dynamic end-coupling. First, we observed fluorescently labeled GMPCPP-stabilized MT seeds gliding along beads coated with Ndc80 and CENP-E, as in the assay with stabilized MTs. Then, unlabeled soluble tubulin was added, so that the motions of the labeled MT segments could be seen clearly (Fig. 4a, b). About half of these segments moved slowly away from the beads, indicating that unlabeled tubulin was incorporated at the bead-bound plus-ends (Supplementary Movie 6). After several minutes, 83% of these segments rapidly moved back, leading to the saw-toothed distance vs. time plots that are typical for dynamic MTs (Fig. 4c). The velocity of "away" motion (0.67 ± 0.04 μm min$^{-1}$) was similar to the rate of polymerization of the bead-free MT ends (Fig. 4d). This is significantly slower than CENP-E-dependent MT gliding, suggesting that the away motion did not result from lateral CENP-E-dependent gliding. We varied the concentration of soluble tubulin

or MgCl$_2$, a known effector of MT polymerization[28]. The velocity of away motion changed along with the rate of free MT end polymerization (Supplementary Fig. 6, Supplementary Movie 7), strongly implying that tubulin assembly was indeed the driving force.

To directly determine whether CENP-E was required during the tubulin assembly phase, we examined the dynamics of the bead-coupled MT end in this motor's absence. A commonly used inhibitor of CENP-E, GSK-923294, induces strong motor-MT binding[29]; consequently, we could not use it in our assay because it would firmly attach MTs to the CENP-E beads. Instead, we took advantage of rare events of direct MT end-binding by chance encounters to beads coated with Ndc80 alone. About 3% of MT segments ($n ≈ 530$) attached via their ends and moved away from the beads after soluble tubulin was added (Fig. 4e). This velocity was very similar to that observed on beads coated with both Ndc80 and CENP-E (Fig. 4d), consistent with our interpretation that it was determined by tubulin assembly. Strikingly, most of the MT segments that moved back to the Ndc80-coated beads after MT catastrophe failed to initiate new away motion (Fig. 4f, g). Thus, CENP-E motor is required for re-establishing dynamic coupling after each depolymerization phase, explaining why the total duration of dynamic coupling was significantly reduced in its absence (Supplementary Fig. 6c).

Motion of the labeled MT fragment toward Ndc80+CENP-E beads was much faster (18.3 ± 1.3 μm min$^{-1}$) than the away

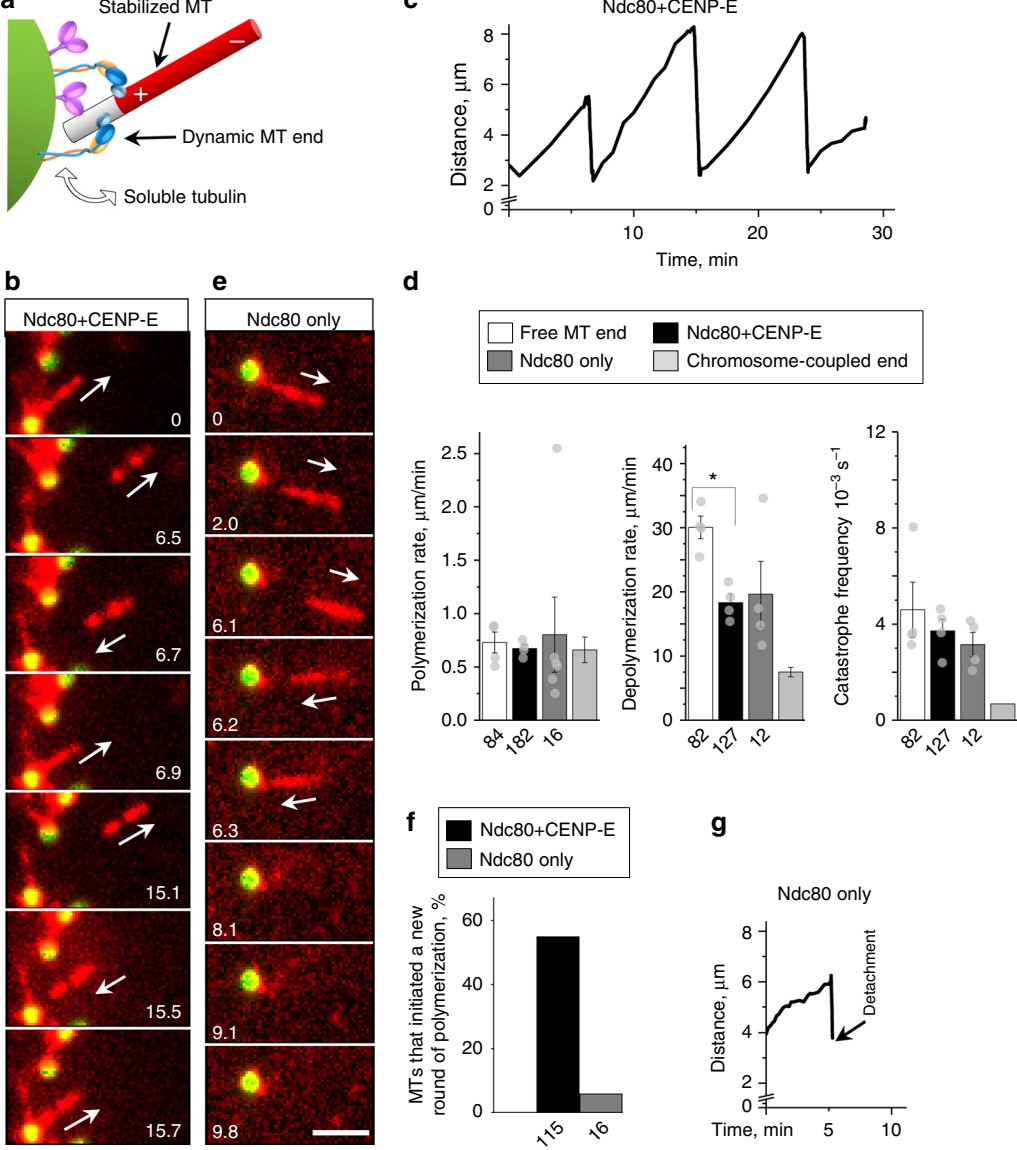

**Fig. 4** Centromere-associated protein E (CENP-E) and Ndc80 coupling to the dynamic microtubule (MT) ends. **a** Schematics of the dynamic MT end-conversion assay. Fluorescently labeled GMPCPP-stabilized MT seeds glide on beads, then unlabeled soluble tubulin is added to examine its incorporation at the bead-bound MT plus-end. **b** Selected time-lapse images recorded with Ndc80+CENP-E beads after addition of unlabeled soluble tubulin (6.3 μM). Numbers are time (min) from the start of observation. Arrows show the direction of motion of the bright MT fragment, reporting on the dynamics of the bead-bound MT plus-end. Bar, 3 μm. **c** Distance from the distal tip of the fluorescent MT fragment to the bead vs. time, showing repeated cycles. **d** Dynamics parameters for freely growing MTs ($N = 4$ independent trials) and for MT ends coupled to protein-coated beads ($N = 4$ for Ndc80+CENP-E; $N = 6$ for Ndc80 alone), showing means ± SEM for average results from these trials, source data are provided as a Source Data file. Data for isolated mammalian chromosomes (chromosome-coupled end) are from ref. [30]. Statistical differences were evaluated by Kruskal–Wallis analysis of variance (ANOVA); *$p < 0.05$. **e** Images as in **b** but recorded using beads coated with Ndc80 only. **f** Percent of bead-coupled MT ends that disassembled, and then initiated a new round of MT polymerization without losing their bead attachment. Source data are provided as a Source Data file. **g** Plot similar to (**c**) but for a bead coated with Ndc80 in the absence of CENP-E motor. The Ndc80-coated beads can maintain coupling only for one dynamic MT cycle, detaching after MT depolymerization

motion. Because our assay lacks any depolymerizing or minus-end-directed motors, it could only be driven by MT depolymerization. Consistently, this velocity, like the catastrophe frequency, was not significantly changed in the absence of CENP-E (Fig. 4d). Interestingly, depolymerization of the bead-coupled MT ends was slower than that of the free MT end. This retardation and the slightly reduced catastrophe frequency are consistent with the previously reported suppression of the dynamics of MT ends coupled to the kinetochores of purified mammalian chromosomes in vitro[30]. Thus, the MT wall-binding proteins CENP-E

and Ndc80 can achieve dynamic end-coupling that is similar to physiological.

**End-coupling by other MAPs is not as lasting as with Ndc80.** We applied these experimental and modeling tools to investigate the molecular requirements for end-conversion in vitro. The essence of our model for the CENP-E–dependent MT wall-to-end transition is that the motor positions Ndc80 molecules near the MT plus-end, where they provide dynamic adhesion via MT

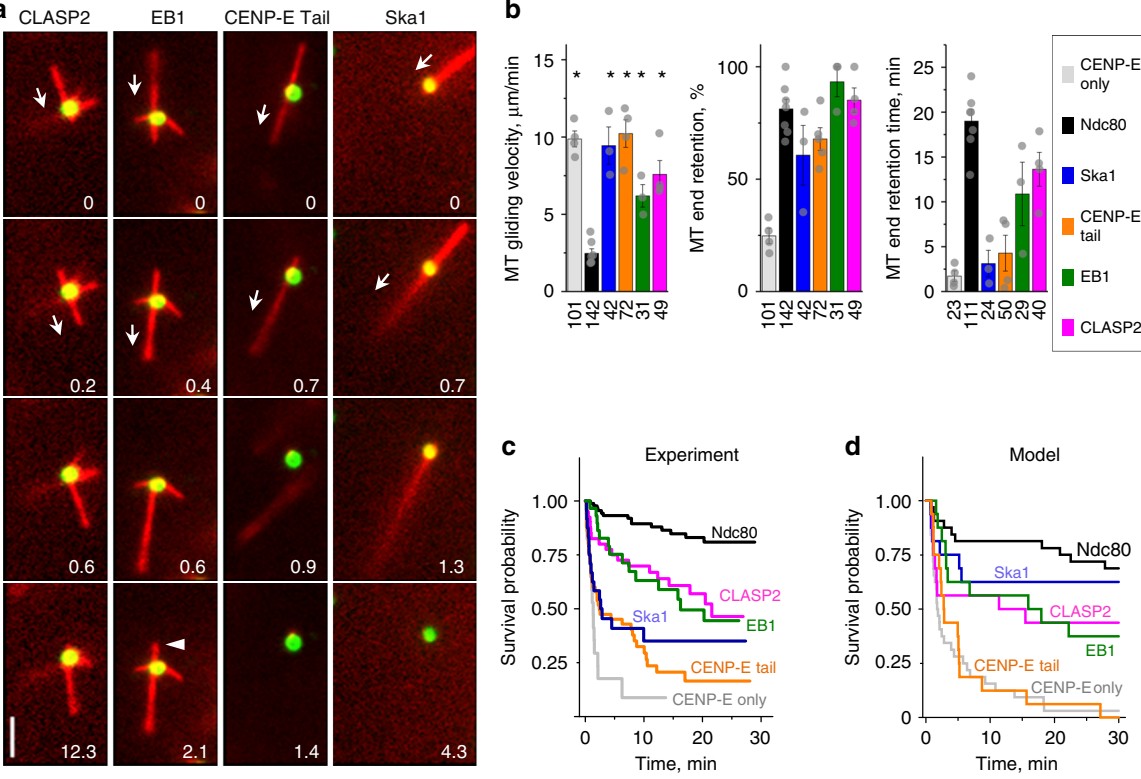

**Fig. 5** Microtubule (MT) wall-to-end transition in molecular systems combining CENP-E with various MT-associated proteins (MAPs). **a** Selected time-lapse images of stabilized MTs moving over beads coated with CENP-E motor and the indicated MAP. All proteins were conjugated to beads via anti-GFP antibodies to achieve similar brightnesses, ensuring that any differences in MT interactions are not due to differences in the density of the protein coatings. Numbers are time (min). Arrows show the direction of MT gliding. Bar, 3 μm. Arrowhead in the last EB1 panel points to a loss of tip attachment due to the MT end-to-wall transition. **b** Quantifications as in Fig. 2c, but for beads coated with mixtures containing the CENP-E motor and the indicated MAP. Data are means ± SEM for results from $N$ independent trials, which are shown with gray dots. For CENP-E only, CENP-E paired with either Ndc80 Broccoli, Ska1 complex, CENP-E Tail, EB1, or CLASP2 $N = 4, 7, 3, 4, 3,$ and 4, respectively. Total number of observed events is indicated below each column. Asterisk above a bar ($p < 0.05$) indicates a significant difference relative to the analogous measurement for Ndc80 beads, as determined by Kruskal–Wallis analysis of variance (ANOVA). Source data are provided as a Source Data file. **c** Kaplan–Meier plot for MT end-retention time based on the same data sets as in (**b**). **d** Kaplan–Meier plot for the predicted MT end-retention time for different MAPs paired with CENP-E motor ($n = 32$ simulations for each condition). Different MAPs were modeled using the diffusion coefficients and residence times measured for single molecules in vitro (Supplementary Table 2)

wall-diffusion. If this model is accurate, other diffusing MAPs could substitute for the Ndc80 function. To test this prediction, we used four proteins that can diffuse on the MT wall: Ska1 complex[12], the MT-binding tail of CENP-E[19] (CENP-E Tail), EB1[31,32], and CLASP2[33]. Purified green fluorescent protein (GFP)-tagged versions of each of these proteins were combined with CENP-E motor. CENP-E transported stabilized MTs quickly over these beads, implying that these MAPs imposed less frictional resistance than Ndc80 (Fig. 5a, b). With all these MAPs, we observed that a large fraction of MT plus-ends reached the protein-coated beads and stayed attached for at least 4 s, indicating that these molecular pairs are capable of wall-to-end transition. However, the average duration of end-attachments differed among the MAPs tested: Ska1 complex (Supplementary Movie 8) and CENP-E Tail maintained end-attachments for only 3–4 min, whereas with EB1 and CLASP2, >80% of end-attachments lasted 11–14 min (Fig. 5b, c).

To identify the determinants of these differences in end-retention, we asked whether our mathematical model could recapitulate these findings based on the quantitative characteristics of these MAPs, such as the rate of MAP single-molecule diffusion and the duration of one such interaction (Fig. 3c). We used single-molecule visualizations in vitro to measure these parameters for Ndc80, Ska1, and CLASP2, and

used published results for EB1 and CENP-E Tail (Supplementary Fig. 7a–c). Based on these input parameters (Supplementary Table 2), the model correctly describes that Ndc80 has the longest attachment time, CLASP2, EB1, and Ska1 exhibit intermediate end-retention, and CENP-E Tail has the shortest (Fig. 5d).

Another consistency between our model and experiment is the propensity of MTs to diffuse rapidly on EB1-coated beads. Although EB1 supported wall-to-end transition, the resultant end-attachments in vitro were often interrupted by the reverse end-to-wall transitions (29% from $n = 31$ coupled MTs, Fig. 5a, arrowhead). During these events, the MT end lost its bead association and MT diffused vigorously on the bead (Supplementary Movie 9). This behavior is consistent with the fast diffusion rate and short residence time for EB1 (Supplementary Table 2), which mediates MT diffusion when all CENP-E motors accidentally unbind. By contrast, we did not observe such reverse end-to-wall transitions with Ndc80, as explained in the model by this protein's relatively slow diffusion and longer residence time (Supplementary Fig. 7d).

We investigated coupling of dynamic MT ends by EB1 and CLASP2 proteins (Fig. 6), which had moderately long end-retention with stabilized MT ends. With dynamic MTs and EB1+ CENP-E beads, there were repeated wall-to-end and end-to-wall transitions interspersed with directed transport (Fig. 7), so the

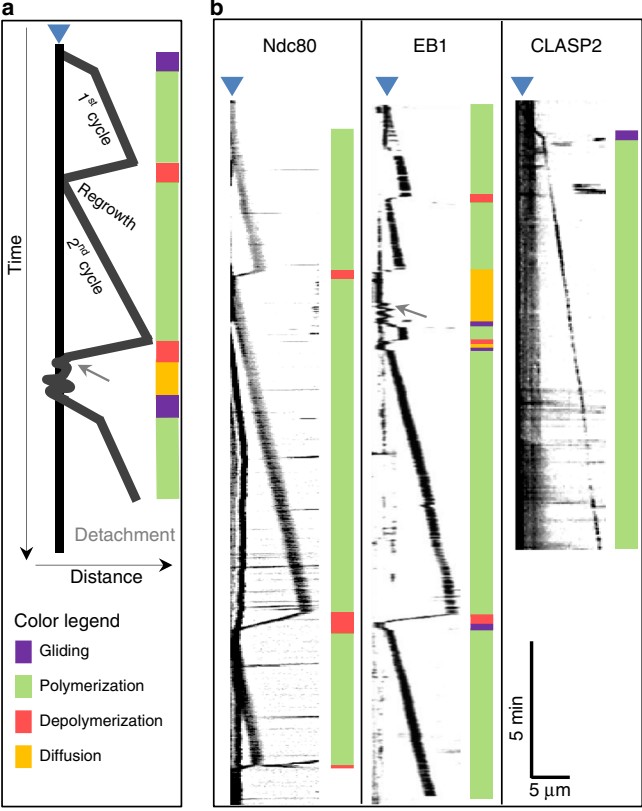

**Fig. 6** Behavior of dynamic microtubules (MTs) during end-conversion in vitro. **a** Schematics depicting different features of experimental kymographs. Fluorescent MT segments move away or toward the bead owing to the addition or loss of unlabeled tubulin from the bead-coupled MT plus-end, motor-dependent transport (gliding), and MT diffusion on bead surface. Vertical black line corresponds to a coverslip-immobilized bead (denoted by blue triangle above), which is often visible in the MT channel owing to the bead-attached motionless MTs. When such MTs were lacking, bead position was determined from green fluorescent protein (GFP) channel. The oblique black lines correspond to motions of the brightly labeled GMPCPP-stabilized MT seeds. Color bars on the right provide visual guides for interpretations of these motions. Arrowhead represents MT detachment; arrow represents end-to-wall transition event leading to MT diffusion. **b** Example kymographs for dynamic MT ends coupled to beads coated with indicated proteins together with CENP-E

total end-retention time for EB1 (Fig. 5b, c) is likely to be overestimated. Interestingly, in the presence of soluble tubulin, the CLASP2+CENP-E pair could support the away motions and had no reverse end-to-wall transitions, but the backward depolymerization-associated motions were not seen (Fig. 6). Thus, although CLASP2 mediates moderately long attachments of stabilized MT ends, unlike Ndc80 it fails to couple to the ends of depolymerizing MTs.

**Kinesin-1 cannot replace CENP-E during MT end-retention**. To further investigate biophysical mechanisms of end-conversion, we tried to replace CENP-E with another transporting plus-end-directed motor, Kinesin-1. In the absence of Ndc80, stabilized MTs glided much faster on the Kinesin-1 beads than on beads coated with CENP-E (Fig. 8a, b), consistent with the higher velocity of Kinesin-1[34]. Upon addition of Ndc80, the MT gliding slowed down, and the fraction of MT plus-ends that paused at the

beads for >4 s also increased. However, most of these ends detached quickly at the same Ndc80 concentrations that provided durable attachments to the ends delivered by CENP-E (Supplementary Movie 10). MT end-retention time by Ndc80 in the presence of Kinesin-1 was much lower than in the presence of CENP-E (Fig. 8c) or the end-retention time of dynamic MTs by Ndc80 alone (Supplementary Fig. 6c). Thus, Kinesin-1 can efficiently deliver the MT plus-end to Ndc80, but it subsequently actively interferes with Ndc80-mediated MT tip attachment.

To obtain mechanistic insight into this unexpected result, we turned to our mathematical model, in which motor function is described using two force-dependent characteristics, for velocity and unbinding (Fig. 3c). Although CENP-E was initially thought of as a mitotic version of Kinesin-1[35], the motility characteristics of the two motors under force are markedly different (Fig. 9a)[20,35,36]. When the force-velocity and force-unbinding Kinesin-1 characteristics were incorporated, the model correctly predicted a low survival probability for MT-bead attachments (Fig. 9b). We used this tool to find conditions that "convert" Kinesin-1 into CENP-E, and discovered that end-retention could be extended in silico using Kinesin-1 characteristics corresponding to 100-fold lower ATP concentration. Under these conditions, Kinesin-1 walks much slower and also has reduced MT unbinding rate[37] (Fig. 9a; Supplementary Note 1). We tested this prediction experimentally. At 20 μM ATP, the MT gliding velocity on Ndc80+ Kinesin-1 beads decreased (Fig. 8c), as expected. Consistent with the model prediction, the arriving MT ends remained attached to these beads for much longer (Fig. 9b). The distinct behaviors of CENP-E and Kinesin-1 motors in the MT wall-to-end transition assay and the effect of low ATP concentration were further revealed by plotting the end-retention times of individual MTs against their preceding gliding velocities (Fig. 9c). Although these data were highly variable, the points for CENP-E and Kinesin-1 exhibited minimal overlap at 2 mM ATP, but under low ATP conditions for Kinesin-1 they became highly similar.

To uncouple the impact of velocity retardation from more stable MT association, both of which are induced by low ATP concentration in experimental assay with Kinesin-1, we modeled these force-dependent characteristics separately. Significantly longer MT end-retention time could be obtained in the model using the unbinding rate of CENP-E in combination with the naturally high velocity of Kinesin-1 but not vice versa (Fig. 9d). Thus, the distinct end-retention behavior of CENP-E and Kinesin-1 motors stems from their different unbinding rates under force.

## Discussion

MT end-conversion by mitotic kinetochores is one of the least understood transition that occurs during chromosome segregation. Until now it has not been reconstructed and analyzed using quantitative approaches. Previous experiments using Ndc80-coated beads mimicked MT end-conversion by triggering depolymerization of the laterally attached MT, causing the bead to follow the depolymerizing tip[17]. Other studies used laser beams to promote direct binding between MT tips and Ndc80 beads, which subsequently remained attached to the dynamic ends for few minutes[13]. It remains unclear whether these events reflect natural end-conversion, especially because MAP-coated beads can follow MT tips while rolling[38]. The approach described here recreates a more physiologically relevant situation by introducing immobilized beads coated with motor domains of the plus-end-directed CENP-E kinesin, which during chromosome congression transports the Ndc80-containing kinetochores.

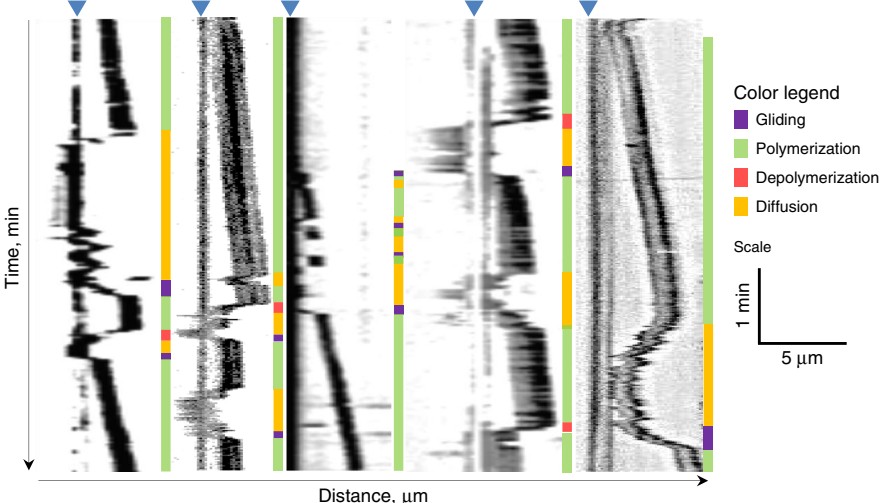

**Fig. 7** Frequent loss of microtubule (MT) end-coupling by EB1 and CENP-E. Five typical kymographs of dynamic MTs coupled to beads coated with a mixture of EB1 and CENP-E proteins. Shown are enlargements of experimental kymographs illustrating highly complex behavior of EB1-dependent MT end-coupling, which is lost and reestablished frequently. MT elongation continued during MT diffusion on the bead, leading to "gaps" in typical MT polymerization kymograph pattern. Colored bars provide interpretations for the kymographs. For other details, see legend to Fig. 6

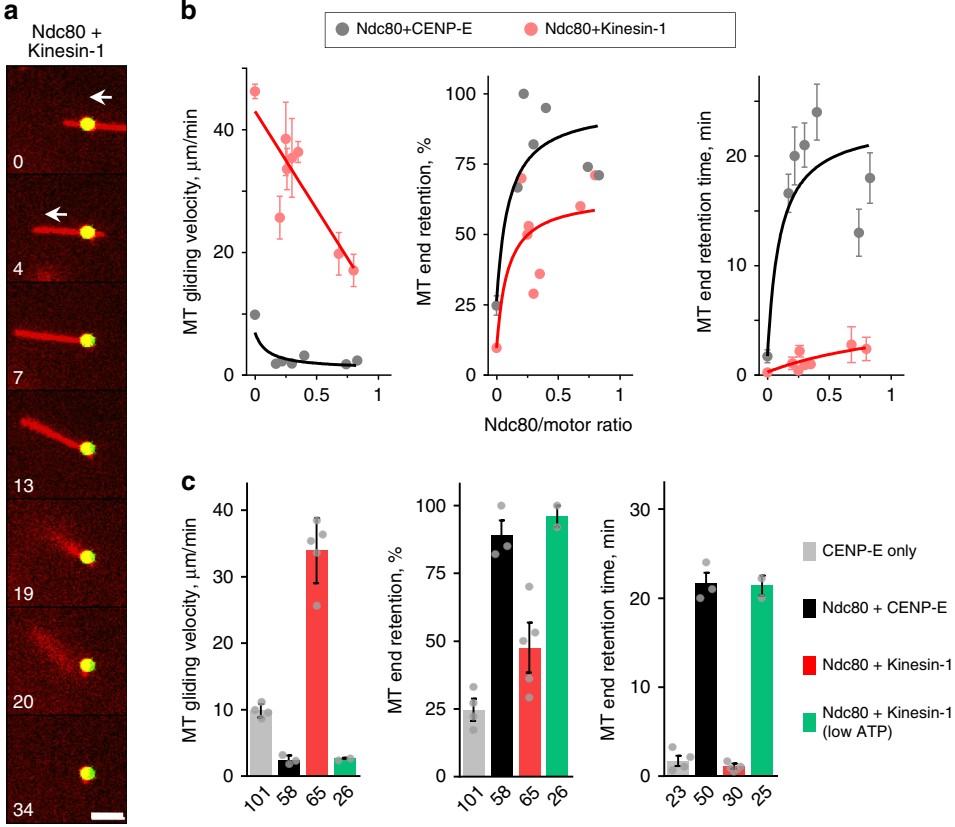

**Fig. 8** Kinesin-1 paired with Ndc80 in the microtubule (MT) wall-to-end transition assay. **a** Selected time-lapse images of a stabilized MT moving over a bead coated with Kinesin-1 and Ndc80 proteins in experiment carried out as in Fig. 2a. Numbers are time (s). Arrows show direction of MT gliding. Bar, 3 μm. **b** MT behavior as a function of the ratio of bead-bound proteins. Points are means ± SEM; curves are exponents (left) and hyperbolic functions illustrating the trends. Source data for (**b**, **c**) are provided as a Source Data file. **c** Average results for beads coated with Ndc80 and motors at ratios from 0.2 to 0.5. Bars show means ± SD for results from *N* independent trials, which are shown with gray dots. For CENP-E only, Ndc80 paired either with CENP-E, Kinesin-1, or Kinesin-1 at low adenosine triphosphate (ATP), N = 4, 3, 5, and 2, respectively. Numbers under each bar indicate the total number of observed events. Experiments were carried out in motility buffer supplemented with 2 mM ATP, except at low ATP concentration which was 20 μM

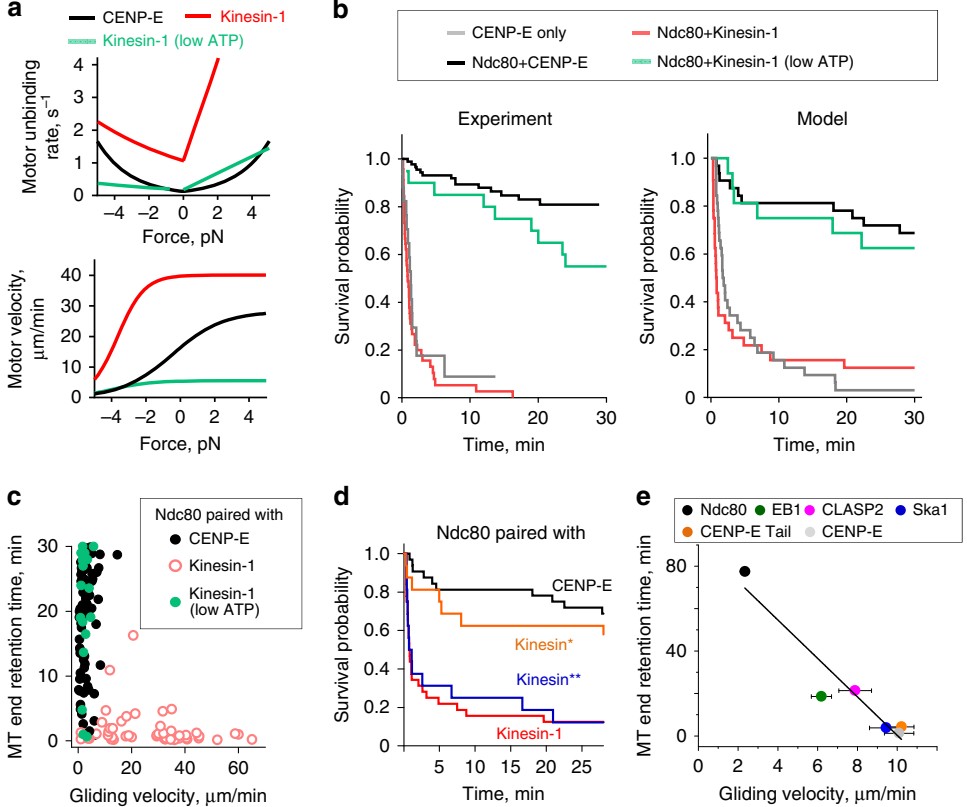

**Fig. 9** Impact of motor's force-sensitivity on microtubule (MT) end-retention. **a** Force-unbinding (top) and force-velocity (bottom) characteristics used in model simulations, see Supplementary Note 1 for details. **b** Kaplan–Meier plots for the MT end-retention time. Experimental results are as in Fig. 2d but supplemented with measurements for Ndc80+Kinesin-1 at 2 mM adenosine triphosphate (ATP) ($N = 7$, $n = 50$) and 20 μM ATP ($N = 2$, $n = 25$). Theoretical plot is based on $n = 32$ simulations for each condition modeled using characteristics in **a**. **c** MT end-retention time vs. preceding gliding velocity for individual MTs in experiments with stabilized MTs. Data are based on $N = 5$, $n = 76$ for Ndc80+CENP-E; $N = 7$, $n = 50$ for Ndc80+Kinesin-1 (2 mM ATP); $N = 2$, $n = 24$ for Ndc80+Kinesin-1 in low ATP (20 μM ATP). **d** Kaplan–Meier plots calculated for molecular patches containing Ndc80 molecules and motor molecules with different force-dependent characteristics. Predictions for Ndc80+CENP-E and Ndc80+Kinesin-1 are the same as in **b**. Prediction for Kinesin* was calculated using force-dependent unbinding rate of CENP-E and the force-velocity function of Kinesin-1. Prediction for Kinesin** was calculated using the force-dependent unbinding rate of Kinesin-1 and the force-velocity function of CENP-E. **e** Average duration of MT end-retention vs. average gliding velocity in MT wall-to-end transition assay using CENP-E motor and indicated MT-associated proteins (MAPs). Duration of MT end-attachment is represented by the half-life of an exponential fit to the corresponding survival probability curve in Fig. 5c. Horizontal error bars are same as for MT end-retention time in Fig. 5b. Black line is the linear fit to all points. Pearson's correlation analysis gives $R^2 = 0.91$ with 95% confidence, and hence the anti-correlation with gliding velocity is significant

Our study reveals that the pair of only two proteins, CENP-E motor and Ndc80 complex, can achieve highly efficient MT end-conversion in vitro. During lateral transport, Ndc80 binding to the MT wall creates drag that antagonizes the walking of the CENP-E motor, suggesting that the 10-fold lower velocity of chromosome congression relative to freely walking CENP-E[39,19,35] is a result of molecular friction from kinetochore-bound proteins, such as Ndc80 complexes. Near the MT plus-end, CENP-E and Ndc80 generate a physiologically competent MT attachment, as seen from its ability to persist during several dynamic cycles of the coupled MT end, lasting tens of minutes.

Several lines of evidence suggest that the underlying molecular interactions occur at the wall of the MT near its tip, rather than requiring these proteins to bind specifically to the protruding protofilaments, which are more curved than protofilaments within intact MT walls[17,40]. First, both CENP-E motors and Ndc80 are MT wall-binders with no intrinsic ability to bind MT tips in vitro[12,16,19]. Second, these molecules are immobilized randomly on the bead surface, rather than clustered in one spot; therefore, the most likely configuration is lateral attachment of these molecules along the MT wall segment immediately adjacent to the tip, as in our model (Supplementary Movie 5). Third, end-

conversion is not a unique property of these two proteins. Other MAPs can substitute to some extent for Ndc80 in vitro, but there is no clear correlation between their performance in this assay and their MT tip-binding properties. For example, EB1 strongly prefers growing, but not depolymerizing, MT tips[31], whereas the Ska1 complex can track both growing and shortening MT ends[12,41]. Both proteins, however, were less effective than Ndc80 in the end-conversion assay. Fourth, the duration of MT end-retention correlates with the velocity at which CENP-E transports MTs laterally bound to these MAPs (Fig. 9e), suggesting that maintenance of end-attachment is determined mostly by the MT wall-binding properties of these MAPs. Fifth, the survival plots for the MT end-retention time for these MAPs were similar to those obtained in the model using only two input parameters, the diffusion rate and residence time (Fig. 5c, d), both of which describe MT wall-binding. Thus, the model suggests that Ndc80 provides the best end-retention among the tested MAPs due to a combination of slow diffusion and relatively long residence time on the MT wall.

There is impressive quantitative consistency between our experimental observations and the model's prediction of end-conversion with Kinesin-1, suggesting that our interpretations are

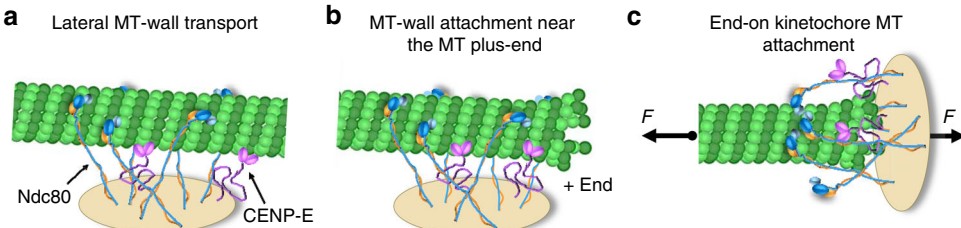

**Fig. 10** Key phases of the CENP-E-dependent microtubule (MT) end-conversion in cells. Schematics show multimolecular ensemble of CENP-E kinesins and Ndc80 complexes, forming a molecular lawn that interacts with the MT. **a** Ndc80 slows down CENP-E kinesin during the plus-end–directed transport. Ndc80 plays an essential role in providing durable and mobile attachment to the end-proximal MT wall. In our in vitro experiments and in silico, these molecular interactions are concentrated at one side of the MT, which forms oblique contact with the molecular lawn. This configuration is also likely to occur transiently at the kinetochores of mitotic cells, as shown in **b**. However, forces acting on the chromosomes and kinetochore-bound MTs reorient the kinetochore, promoting the classical end-on configuration (**c**). We propose that in this configuration, kinetochore attachment is mediated by essentially the same molecular interactions with the MT wall, as described in this work

mechanistically accurate. At physiological ATP concentration, Kinesin-1 proficiently delivers the MT plus-ends to the bead-conjugated Ndc80 molecules, but in its presence the Ndc80 molecules fail to provide lasting attachment to these ends. Reducing the ATP concentration dramatically increases the end-retention time for Kinesin-1 in experiments and in our model. Low ATP reduces both the velocity of Kinesin-1 transport and its unbinding rate from the MT wall under force, and these characteristics become similar to those of CENP-E. Using modeling, we show that only the latter dependency is essential for MT end-retention. Therefore, CENP-E is a specialized transporting motor with a force-dependent unbinding rate that is well suited for cooperation with the Ndc80 molecules during end-conversion.

Interestingly, the retardation of CENP-E transport by Ndc80 predicted by the model was not as strong as that observed in experiment. This suggests that Ndc80–MT wall motility under force may differ from that of other MAPs, an important prediction that should be tested in future work. The velocity retardation, however, is not sufficient for successful end-retention, as Ndc80 mutant lacking N-terminal tail of Hec1 subunit slowed down MT transport but could not provide durable end-retention (Fig. 2c). Another difference between theory and experiment is that Ska1 protein has a shorter end-retention time in vitro than predicted by the model. This could be explained by some aspect of the experimental system, such as suboptimal activity of bead-immobilized Ska1 due to the specific location of the tag. Alternatively, this discrepancy could indicate that MT-binding characteristics that are not yet included in our model, or are assumed to be equivalent for all MAPs, also influence end-retention time.

Together, our data argue that motility characteristics of the motors and MAPs forming a multimolecular ensemble must be finely tuned to enable emergent end-retention behavior. These molecules maintain a short (20–40 nm) overlap with the distal MT wall segment, similar to the overlap created by the multimolecular "sleeve" in Hill's model[42], although the underlying biophysical mechanisms are different[10]. Because the stepping of the MAPs forming the sleeve is highly coordinated, it tends to engage maximal number of MAPs, corresponding to the minimal free energy of this system. In our model, all molecular interactions are uncoordinated and stochastic, as proposed in the kinetochore "molecular lawn" representation[43]. Here, entropic forces play a significant role, and the number of MT-bound molecules is limited by kinetic binding constants[44]. These factors, together with the force-sensitivity of stepping and unbinding, determine the outcome of end-retention by an ensemble of motors and MAPs. We note that at a single-molecule level, this mechanism is highly similar to the proposed "tethered motor" model for MT tip tracking by full-length CENP-E[19].

Our model for an ensemble behavior of the stochastic and force-sensitive MT wall-binding proteins lays a conceptual foundation for understanding MT end-conversion during mitosis. First, our work suggests that CENP-E–mediated end-conversion in cells can take place without specialized tip-binding proteins or regulatory modifications, which may nonetheless provide additional layers of complexity. Second, these reconstructions provide important insights into the formation of end-attachment in cells, which we propose to require two distinct driving factors: molecular MT wall interactions, as reconstituted in this work, and external forces that induce end-on geometry for kinetochore–MT attachments in cells but are absent in our current system. In this view, the initial MT-lateral attachment of the kinetochores is first replaced by binding to the terminal MT end segment lying obliquely to the kinetochore molecular lawn (Fig. 10a, b). The MT-perpendicular end-on configuration is then induced by spindle forces that orient sister kinetochores along the spindle axis (Fig. 10c). We propose that despite these different MT-kinetochore orientations, the nature of the underlying molecular interactions remains the same: in the MT-perpendicular end-on configuration, Ndc80 and CENP-E continue to interact with the tip-adjacent MT wall via essentially same biophysical mechanism as in the oblique configuration. Thus, this mechanistic model provides a consistent and unifying molecular basis for the initial lateral transport, wall-to-end transition, and durable end-coupling at vertebrate kinetochores.

## Methods

**Protein purification.** Tubulin from bovine brain was purified by thermal cycling and chromatography and labeled with HiLyte647 as in ref. [45]. The *Xenopus laevis* CENP-E-GFP construct (1–473 amino acids (aa)) was expressed and purified from *Escherichia coli* as in ref. [46]. This truncated CENP-E protein contains the motor domains of CENP-E dimerized with a short segment of native stalk, but the rest of the stalk and the MT-binding tails are absent (Supplementary Fig. 1a). Truncated CENP-E with C-terminal Myc-tag in place of GFP was a gift from Drs. Y. Kim and D. W. Cleveland. The GFP-tagged C-terminus of CENP-E (CENP-E Tail) was purified as in ref. [19]; human Bonsai Ndc80-GFP protein complex as in ref. [26]; human Broccoli Ndc80-GFP construct as in ref. [12]; full-length human EB1-GFP as in ref. [47]; GFP-tagged Kinesin-1 (1–560 aa) as in ref. [48]; and full-length human Ska1-GFP complex consisting of all three subunits Ska1, Ska2, and Ska3 as in ref. [49]. Human GFP-CLASP2 was purified using Baculovirus Expression Vector System (BD Biosciences, San Jose, CA, USA). Briefly, CLASP2 ORF was subcloned into expression vector pKL in fusion with an N-terminal GFP-tag and a C-terminal His10-tag. The resultant plasmid, pKL-GFP-CLASP2-His10x, was used to transform DH10EmBacY *E. coli* for transposition into bacmid. Production of recombinant baculovirus and transfection of *Spodoptera frugiperda* Sf21 cells was performed using the MultiBac expression system[50]. Sf21 cells were lysed by sonication in 50 mM Tris-HCl pH 8.0, 150 mM NaCl, 7 mM β-mercaptoethanol supplemented with protease inhibitors (Complete EDTA-free, Roche, Basel, Switzerland). Protein extracts clarified by centrifugation at 20,000 × *g* for 30 min were loaded onto a HisTrap HP column (GE Healthcare, Chicago, IL, USA) pre-equilibrated in 50 mM Tris-HCl pH 8.0, 500 mM NaCl, 20 mM imidazole, 7 mM

β-mercaptoethanol, and eluted with 200 mM imidazole. CLASP2-containing fractions were pooled and further purified on a HiPrep 16/60 Sephacryl S-300 HR column (GE Healthcare) pre-equilibrated with 50 mM sodium phosphate buffer pH 7.0, 400 mM KCl, 2 mM MgCl$_2$, 0.8 mM β-mercaptoethanol, and 1% glycerol.

**Fluorescence microscopy assays.** Prior to each motility experiment, a frozen protein aliquot was thawed and clarified by ultracentrifugation (TLA100 rotor, Beckman Coulter, Brea, CA, USA) at 156,845 × $g$ for 15 min at 4 °C. Protein concentration in the supernatant was determined by measuring GFP intensity by fluorescence microscopy and comparing to a "standard" GFP-labeled protein whose concentration was determined by spectrometry[51]. For gliding assays, taxol-stabilized MTs were prepared from unlabeled and HiLyte647-labeled tubulin in a 24:1 ratio. For MT wall-to-end transition assays, stabilized MTs were prepared from a mixture of unlabeled and HiLyte647-labeled tubulin (10:1, total tubulin concentration, 72.5 µM) and 1 mM GMPCPP (Jena Bioscience, Jena, Germany) incubated at 37 °C for 10 min. All motility assays were carried out using a Nikon Eclipse Ti-E inverted microscope equipped with 1.49× NA 100× oil objective and Andor iXon3 CCD camera (Cambridge Scientific, Watertown, MA, USA), as in ref. [19]. Under these conditions, the microscope produced 512 × 512-pixel images with 0.14 µm pixel$^{-1}$ resolution in both the $x$ and $y$ directions. All experiments were carried out at 32 °C by heating the objective with an objective heater (Bioptechs, Butler, PA, USA).

**Motility assays using free-floating beads and a laser trap.** COOH-activated glass beads (0.5 µm, Bangs Laboratories, Fishers, IN, USA) were coated with a mixture of Broccoli Ndc80-GFP and truncated CENP-E-GFP using 12 nm DNA links as in ref. [20], then soluble proteins were removed. The ratio of these proteins was varied while holding constant the combined total protein concentration in bead preparation buffer (40 nM). In experiments with dynamic MTs, the Ndc80/CENP-E ratio was 0.25. Because both proteins were conjugated to beads through a GFP-tag, this ratio reflects the proportion of Ndc80 and CENP-E molecules conjugated to the surface of the beads. Perfusion chambers were prepared by attaching a silanized glass coverslip over a regular glass slide with double-sided sticky tape (Scotch) to generate a 15 µl flow chamber; solutions were exchanged using a peristaltic pump as in ref. [52]. Taxol-stabilized MTs were prepared and immobilized on the coverslip using anti-tubulin antibodies (Serotec, Cat. no. MCA2047), used in concentration 10 µg ml$^{-1}$ as in ref. [20]. For experiments with dynamic MTs, digoxigenin (DIG)-labeled GMPCPP-stabilized MT seeds were prepared and immobilized on the coverslip via anti-DIG antibodies (Roche, Cat. no. 11333089001), used in concentration 20 µg ml$^{-1}$ as in ref. [41]. Next, beads were flowed into the chamber in "imaging" buffer: BRB80 (80 mM PIPES, 4 mM MgCl$_2$, 1 mM EGTA, pH 6.9) supplemented with 4 mg ml$^{-1}$ bovine serum albumin (BSA), 2 mM dithiothreitol (DTT), 2 mM Mg-ATP, 6 mg ml$^{-1}$ glucose, 80 µg ml$^{-1}$ catalase, and 0.1 mg ml$^{-1}$ glucose oxidase. Here and in all other motility assays, for experiments with taxol-stabilized MTs, the imaging buffer was further supplemented with 0.5% β-mercaptoethanol and 7.5 µM taxol. For experiments with dynamic MTs, the imaging buffer was supplemented with 6 µM soluble tubulin, 1 mM Mg-GTP and DTT at final concentration of 10 mM. Using the 1064-nm laser beam of our laser trap[36], a free-floating bead was trapped and brought into contact with the wall of the taxol-stabilized MT as described in ref. [20], examining 4–12 beads for each Ndc80/CENP-E ratio. In experiments with dynamic MTs, the beads attached spontaneously to the MTs. Image acquisition in differential interference contrast mode was performed with exposure times of 100 ms using either a Cascade 650 CCD (Photometrics, Tucson, AZ, USA) or an Andor iXon3 camera controlled by the Metamorph software (Molecular Devices, San Jose, CA, USA) or NIS-Elements software (Nikon Instruments, Melville, NY, USA), correspondingly.

**MT gliding assay.** Perfusion chambers and coverslips coated with proteins were prepared as described in ref. [51] using a mixture of 20 µg ml$^{-1}$ biotinylated anti-Myc antibodies (EMD Millipore, Burlington, MA, USA, Cat. no. ab34773) and 20 µg ml$^{-1}$ biotinylated anti-GFP antibodies (Abcam, Cambridge, MA, USA, Cat. no. ab6658) in wash buffer (BRB80 supplemented with 4 mg ml$^{-1}$ BSA, 0.1 mM Mg-ATP, and 2 mM DTT). A solution of 0.1 µM Myc-CENP-E in wash buffer was added for 30 min, and then the chamber was washed and incubated with Bonsai Ndc80-GFP at 10–150 nM for 30 min. TIRF (total internal reflection microscopy) images of five different fields were collected with exposure times of 300 ms using a 488 nm laser. The brightness of Bonsai Ndc80-GFP intensity was calculated by averaging mean intensities of these fields using the ImageJ software[53]. The resulting density of Bonsai Ndc80-GFP coating, as judged by GFP fluorescence, increased linearly with increasing concentration of soluble protein (Supplementary Fig. 1b). Then, taxol-stabilized HiLyte647-labeled MTs were introduced in imaging buffer (described in "Motility assays using free-floating beads and a laser trap" section) supplemented with 10 µM taxol, the chamber was sealed with VALAP (1:1:1 vaseline/lanolin/paraffin), and gliding MT motions were recorded continuously using TIRF mode for 10 min. MTs with approximately linear trajectories were selected, and their velocities were estimated from kymographs.

**MT wall-to-end transition assay.** Perfusion chambers with coverslip-immobilized microbeads were prepared as in ref. [51]. Briefly, streptavidin-coated 0.9 µm polystyrene beads bound to coverslips were coated with biotinylated anti-GFP antibody (20 µg ml$^{-1}$), and subsequently blocked with 100 µM biotinylated polyethylene glycol, resulting in negligible MT binding to bead and coverslip surfaces. A solution of 0.3 µM CENP-E-GFP or Kinesin-1-GFP in wash buffer was added and incubated for 30 min, and several images of beads were collected for subsequent quantifications of GFP intensity, corresponding to the density of motor coating. Images of beads coated with these and other GFP-labeled proteins were acquired with a 488 nm 100 mW diode laser (Coherent, Santa Clara, CA, USA) at 10% power with an exposure time of 300 ms. The chambers were then washed and incubated for 30 min with a GFP-labeled MAP. Unless stated otherwise, the MAP concentrations were as follows: Broccoli Ndc80 0.4 µM; CLASP2 0.4 µM; Ska1 0.2 µM; CENP-E Tail 0.1 µM; EB1 0.4 µM. Images of the beads were collected again to record the increase in GFP intensity. The resultant MAP coatings were similar in density (see Supplementary Fig. 8), as judged by the GFP intensities of the beads. Next, GMPCPP-stabilized MT seeds in warm wash buffer in which ATP was replaced with 0.1 mM AMP-PNP (Sigma-Aldrich) were introduced to promote MT binding to the motor molecules on the beads. Then, imaging buffer with 1 mM ATP was washed in and epifluorescence images of HiLyte647-labeled MTs were collected every 4 s for 30 min with exposure times of 300 ms using a 70 mW 640 nm diode laser (Coherent) at 50% power. Supplementary movies were prepared by merging channels from MT and bead fields using Metamorph. In some experiments, AMP-PNP step was omitted, MTs in imaging buffer with 1 mM ATP were added to the protein-coated beads, and the samples were viewed immediately. The results obtained with and without AMP-PNP were very similar (Supplementary Fig. 3), so they were combined into a single dataset. For experiments with Kinesin-1 at low ATP concentration, the imaging buffer was supplemented with 20 µM Mg-ATP instead of 2 mM Mg-ATP. MT end-conversion (MT wall attachment resulting in coupling to dynamic MT ends) was examined with GMPCPP-stabilized MT seeds as above, except that after these MTs glided for 30 min, unlabeled soluble tubulin (6 µM) in imaging buffer supplemented with 1 mM Mg-GTP and DTT at final concentration 10 mM was added. This tubulin solution (40 µl) was perfused for 2 min with a peristaltic pump at 20 µl min$^{-1}$, and imaging continued at 60 frames per min for an additional 30 min. Note that in this assay, MT images were collected using epifluorescence, rather than TIRF, to increase imaging depth. MTs bind all over the surface of the 1 µm bead, rendering most of bead-bound MTs invisible in TIRF. Because epifluorescence illumination has high background when soluble fluorescently labeled tubulin is used, MT elongation in this assay could only be examined using unlabeled tubulin. Under these conditions, motions of the brightly labeled seeds away and toward the bead could be clearly followed, allowing the unambiguous determination of the site of tubulin addition/loss as the bead-bound MT plus-end.

**Quantitative data analyses for MT wall-to-end transition assay.** The brightness of beads coated with different GFP fusion proteins was measured as the integral intensity of the 3.5 × 3.5 µm area encompassing the bead minus the intensity of the same area at a nearby location with no bead. For beads coated with both CENP-E-GFP and MAP-GFP, MAP-GFP brightness was calculated as the GFP fluorescence intensity of a bead after incubation with the MAP-GFP minus the intensity of the same bead before the MAP-GFP was added; the latter corresponds to the brightness motor-GFP. The ratio of Ndc80/motor (CENP-E or Kinesin-1) brightness was doubled to take into account the fact that each Ndc80 molecule has only one GFP, whereas all other tested MAPs are homodimers.

In the MT wall-to-end transition assays, only MTs that satisfied all of the following criteria were selected for quantitative analysis: the MT should be clearly visible and be in focus, the MT should not simultaneously contact several beads, and the MT must have glided until its trailing end reached the bead. The position of the bead in the kymograph was determined by merging the bead and MT images. A successful MT end-retention event was counted only if the trailing MT end was coupled to the bead for longer than two successive frames (4 s). The percentage of end-retention events was calculated from the total number of trailing MT ends reaching the beads. Total MT end-retention time was calculated from kymographs as the time between the end of MT gliding motion and the loss of MT attachment or the end of imaging. Due to the latter events, average end-retention times were underestimated. To overcome this limitation, survival probabilities for bead-coupled MT ends were plotted using the Kaplan–Meier algorithm implemented with the Origin software (OriginLab, Northampton, MA, USA).

In experiments containing soluble tubulin, suspected MT polymerization–driven away motion was identified when a fluorescent MT seed moved sufficiently far from the bead to leave a visible gap between the fluorescently labeled MT segment and the bead. The fraction of dynamic MT attachments was calculated as the ratio of the number of MTs that exhibited at least one away motion to the total number of MTs bound to beads. Velocities of the away/backward motions, their respective durations, and the maximum lengths of polymerized MTs for each dynamic cycle were estimated from kymographs. The velocities of polymerization and depolymerization motions for free MT ends were calculated in an analogous manner. The total dynamic MT attachment time was calculated as the time between the start and end of all dynamic motions or the end of imaging. MT catastrophe frequency was calculated by dividing the observed number of catastrophes by the sum of the elongation time for all polymerizing MTs.

**Measurement of the MT diffusion on the Ndc80-coated beads**. Experimental chambers and solutions were prepared as for the MT wall-to-end transition assay except that beads were coated only with Ndc80-GFP to achieve same Ndc80-GFP brightness as in the wall-to-end transition assay. GMPCPP-stabilized MTs were perfused into the chamber and imaged for 30 min at 15 frames per min. Only those bead-bound MTs that satisfy following criteria were selected for quantitative analyses: MT should be bound only to a single bead, both MT ends should be in focus and clearly visible, and during diffusion MT should only diffuse laterally attached to the bead away from the tip. One end of each of these MTs was tracked manually using ImageJ to obtain distance vs. time track. Next, for each track, initial position was subtracted, and squared displacement vs. time was calculated for each tracked MT. To obtain mean squared displacement plot (MSD) (Supplementary Fig. 4b), the squared displacements were averaged between all MTs. Diffusion coefficient was determined as a half of the slope of MSD of these MT ends over time.

**Measurement of the dynamics of free MT ends**. Dynamics of MT ends that were not bead-bound were examined using the same assay conditions as for MT ends coupled to beads. Coverslip-immobilized streptavidin beads were incubated with $20\,\mu g\,ml^{-1}$ biotinylated anti-mouse IgG antibody (Jackson Immuno Research, West Grove, PA, USA, Cat. no. 115-035-003) for 15 min, followed by $20\,\mu g\,ml^{-1}$ mouse anti-tubulin antibody (AbD Serotec/Bio-Rad, Hercules, CA, USA, Cat. no. MCA2047) for 15 min. HiLyte647-labeled GMPCPP-stabilized MT seeds in warm wash buffer were allowed to bind to these beads, and dynamic MT segments were generated by adding a mixture of unlabeled and HiLyte-labeled tubulins (15:1 ratio, total tubulin concentration, $6.3\,\mu M$) in imaging buffer supplemented with 1 mM Mg-GTP. Images were recorded in TIRF mode for 30 min with exposure times of 300 ms at 60 frames per min. Data in Fig. 4d for the chromosome-coupled MT end is from ref. [30] for a tubulin concentration of $6.8\,\mu M$; catastrophe frequency for these ends was calculated as the inverse of the average time until catastrophe (Table 1 in ref. [30]).

**Single-molecule TIRF assay and data analysis**. Diffusion of different MAPs on the GMPCPP-stabilized MT seeds labeled with HiLyte647 was examined as in ref. [27] using same imaging buffer as in the MT wall-to-end transition assay. MAP concentrations were selected to achieve single-molecule decoration of MTs: 0.1 nM for Ndc80, 0.15 nM for Ska1, and 0.3 nM for CLASP2. Exposure times $t_{exp}$ were 20 ms for Ndc80 and 10 ms for Ska1 and CLASP2. Laser power was 75% for Ndc80 and 100% for Ska1 and CLASP2. To determine diffusion coefficient for each MAP, MSD was plotted vs. time based on >100 tracks of diffusing molecules (Supplementary Fig. 7b). Residence time was determined by analyzing durations of molecular tracks. All visible tracks were selected by clicking on their first and last points using Mathematica (Wolfram Research). To correct for undercount of short-lived binding events, we excluded events shorter than $8 \cdot t_{exp}$, where $t_{exp}$ is exposure time. These cumulative distributions were fitted with truncated exponential functions: $1 - e^{\frac{x - 8\,t_{exp}}{\tau_{eff}}}$. Because the resultant time $\tau_{eff}$ is shorter than real molecular residence time due to photobleaching, the kinetic rate constant of GFP photobleaching under our typical imaging conditions, $k_{bleach}$, was determined as in ref. [27], and the residence time $\tau$ was calculated using the following expression: $\tau = \frac{\tau_{eff}}{1 - \tau_{eff}\,k_{bleach}}$. In our experiments $k_{bleach}$ was $0.17\,s^{-1}$ for Ndc80 complex (at 75% of laser power) and $0.21\,s^{-1}$ for other proteins (measured at 100% laser power).

**Theoretical modeling**. Our model employs Brownian dynamics simulations of ensemble of MAPs and motors interacting with MT. To describe these stochastic molecular interactions, we used force-dependent characteristics for motors and MAPs stepping and unbinding from the MT, which were determined in vitro or based on prior publications [54–63]. Stochastic simulations [64] were carried out using Mathematica (Wolfram Research). For a complete description of the mathematical model and simulation algorithm, see the Supplementary Note 1.

**Reporting summary**. Further information on experimental design is available in the Nature Research Reporting Summary linked to this article.

## Data availability

Source data for quantifications are provided as a Source Data file. Raw images and time-lapse recordings will be made available upon request.

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

## Acknowledgements

Plasmids and protein purification protocols were generously provided by Drs. I. Cheeseman (Whitehead Institute, MIT), J. DeLuca (Colorado State Univ.), T. Surrey (Francis Crick Inst., UK) and D.W. Cleveland (Ludwig Cancer Research and Univ. of CA at San Diego). We are grateful to Drs. H. Maiato and S. Macedo-Ribeiro (Instituto de Investigação e Inovação em Saúde, Universidade do Porto, Portugal) for providing CLASP2 protein. We also thank Dr. A. Kiyatkin, P.-T. Chen and V. Mustyatsa for help with protein purification, and Grishchuk lab members for discussions. Research reported in this publication was supported by the National Institute of General Medical Sciences of the National Institutes of Health under award number R01GM098389 to E.L.G., and by the American Cancer Society grant RSG-14-018-01-CCG to E.L.G. Theoretical modeling was supported by grant from Russian Science Foundation (16-14-00-224) to F. I.A. A.C.F. acknowledges support by the European Research Council (ERC) under the European Union's Horizon 2020 research and innovation programme (grant agreement No 681443) and FLAD Life Science 2020- award (to H. Maiato). F.I.A. acknowledges support from the Russian Foundation for Basic Research 17-00-00481 and 17-00-00480 to E.L.G.

## Author contributions

M.C., E.V.T. and A.V.Z. performed experiments, A.V.Z., M.G. and F.I.A. carried out mathematical modeling, A.C.F. purified human CLASP2 protein, E.L.G and M.C. designed research, analyzed data, and wrote the paper with input from A.V.Z., E.V.T. and F.I.A.

## Additional information

**Competing interests:** The authors declare no competing interests.

