## [Peer Review File · Nature Communications]

Editorial Note: This manuscript has been previously reviewed at another journal that is not operating a transparent peer review scheme. This document only contains reviewer comments and rebuttal letters for versions considered at Nature Communications .

REVIEWERS' COMMENTS:

Reviewer #1 (Remarks to the Author):

In their revised manuscript, Chakraborty et al. have carefully and thoroughly addressed all reviewers' comments and concerns. As a result, the manuscript has improved significantly, and I fully support its publication.

Reviewer #2 (Remarks to the Author):

Grishchuk and colleagues provide a significantly revised study on microtubule wall-to-end conversion by CENP-E and Ndc80c kinetochore components. The authors have made significant changes to the previous version of this manuscript, changing the organization and including additional data to strengthen the mechanistic insights. I find that these changes have greatly strengthened the study, the inclusion of additional modeling is being used very effectively to describe the particular biophysical characteristics of Ndc80 and CENP-E which make this pair especially effective in making and retaining end attachments. The discussion points out the conceptual insights that the study provides, but also mentions its limitations. The authors have addressed my concerns and I can fully recommend publication in Nature Communications.

Minor remaining points:

- avoid repetition of "ill-understood" in abstract
- line 53 "chromosome"
- line 144: there's an interesting difference between the delta80 Ndc80 and K166E mutants, as both prevent effective end binding, but only K166E speeds up CENP-E dependent MT gliding (Figure S2D). The authors should comment on that observation.

Microtubule end conversion mediated by motors and diffusing proteins with no intrinsic microtubule end-binding activity

By Chakraborty et al.

Reviewer 1 had no critical comments.

Reviewer #1 (Remarks to the Author):

In their revised manuscript, Chakraborty et al. have carefully and thoroughly addressed all reviewers' comments and concerns. As a result, the manuscript has improved significantly, and I fully support its publication.

We addressed all minor points raised by reviewer 2:

Reviewer #2 (Remarks to the Author):

Grishchuk and colleagues provide a significantly revised study on microtubule wall-to-end conversion by CENP-E and Ndc80c kinetochore components. The authors have made significant changes to the previous version of this manuscript, changing the organization and including additional data to strengthen the mechanistic insights. I find that these changes have greatly strengthened the study, the inclusion of additional modeling is being used very effectively to describe the particular biophysical characteristics of Ndc80 and CENP-E which make this pair especially effective in making and retaining end attachments. The discussion points out the conceptual insights that the study provides, but also mentions its limitations. The authors have addressed my concerns and I can fully recommend publication in Nature Communications.

Minor remaining points:

- avoid repetition of "ill-understood" in abstract

Replaced with "elusive"

- line 53 "chromosome"

corrected

- line 144: there's an interesting difference between the delta80 Ndc80 and K166E mutants, as both prevent effective end binding, but only K166E speeds up CENP-E dependent MT gliding (Figure S2D). The authors should comment on that observation.

We now discuss this result in Discussion section (in track changes mode)